# Contrail altitude estimation using GOES-16 ABI data and deep learning

Vincent R. Meijer[1,a], Sebastian D. Eastham[1,2,b], Ian A. Waitz[1], and Steven R.H. Barrett[1,c]

[1]Laboratory for Aviation and the Environment, Massachusetts Institute of Technology, Cambridge, USA
[2]Joint Program on the Science and Policy of Global Change, Massachusetts Institute of Technology, Cambridge, USA
[a]Now at: Faculty of Aerospace Engineering, Delft University of Technology, Delft, the Netherlands
[b]Now at: Brahmal Vasudevan Institute for Sustainable Aviation, Department of Aeronautics, Imperial College London, London, UK
[c]Now at: Department of Engineering, University of Cambridge, Cambridge, UK

**Correspondence:** Vincent R. Meijer (vrmeijer@tudelft.nl)

**Abstract.** The climate impact of persistent aircraft contrails is currently estimated to be comparable to that due to aviation-emitted $CO_2$. A potential near-term and low-cost mitigation option is contrail avoidance, which involves re-routing aircraft around ice supersaturated regions, preventing the formation of persistent contrails. Current forecasting methods for these regions of ice supersaturation have been found to be inaccurate when compared to in situ measurements. Further assessment and improvements of the quality of these predictions can be realized by comparison with observations of persistent contrails, such as those found in satellite imagery. In order to further enable comparison between these observations and contrail predictions, we develop a deep learning algorithm to estimate contrail altitudes based on GOES-16 ABI infrared imagery. This algorithm is trained using a dataset of 3267 contrails found within CALIOP LIDAR data and achieves a root mean square error (RMSE) of 570 m. The altitude estimation algorithm outputs probability distributions for the contrail top altitude in order to represent predictive uncertainty. The 95% confidence intervals constructed using these distributions, which are shown to contain approximately 95% of the contrail data points, are found to be 2.2 km thick on average. These intervals are found to be 34.1% smaller than the 95% confidence intervals constructed using flight altitude information alone, which are 3.3 km thick on average. Furthermore, we show that the contrail altitude estimates are consistent in time and, in combination with contrail detections, can be used to observe the persistence and three-dimensional evolution of contrail forming regions from satellite images alone.

## 1 Introduction

Current estimates attribute more than half of aviation's climate impact to contrails (Lee et al., 2020), the ice clouds that form behind aircraft due to mixing of the engine exhaust with the ambient air. Most contrails disappear within minutes after their formation, but when the ambient air is supersaturated with respect to ice they can persist for several hours (Kärcher, 2018). Several options for mitigation of the climate impact of contrails have been proposed, such as contrail avoidance and adopting fuels with lower soot particle emissions (Teoh et al., 2020, 2022; Burkhardt et al., 2018; Voigt et al., 2021). Contrail avoidance involves small changes in aircraft altitude ($\pm$ 2000 ft) to fly around ice supersaturated regions (Sausen et al., 2023),

which have been shown to be horizontally wide but vertically thin (Gierens and Spichtinger, 2000; Spichtinger et al., 2003a). The associated trade-off between additional fuel burn and avoided contrail climate impact has been quantified in modeling studies, indicating that fuel burn penalties smaller than 1% (for re-routed flights) would suffice for halving the radiative impact of contrails (Teoh et al., 2020). These studies assume that the regions to avoid are predicted accurately by the weather data, whereas recent investigations find that the skill of existing numerical weather prediction models in capturing ice supersaturation is limited (Gierens et al., 2020a; Agarwal et al., 2022; Sausen et al., 2023; Geraedts et al., 2023). Inaccurate forecasts may lead to unnecessary deviations (and potentially fuel burn) as well as additional climate impact. As such, the evaluation and improvement of contrail forecasting methods is directly relevant to assessment and implementation of the contrail avoidance concept.

Predictions of contrail formation, persistence and evolution can be evaluated and improved using in situ and remote sensing measurements (Schumann et al., 2017). In situ measurements can provide information on the microphysical properties of a contrail at a great level of detail, whereas remote sensing allows for observations at high temporal frequency and large spatial coverage. The introduction of contrail detection methods for infrared imagery captured by geostationary satellites constitutes one opportunity to obtain large amounts of data that can be used for the evaluation of contrail predictions (Meijer et al., 2022; Ng et al., 2023; Geraedts et al., 2023). However, existing contrail detection methods do not directly provide estimates of the contrail altitude. Given the significance of the vertical position of ice supersaturation, and thus contrails, in the context of contrail avoidance, knowledge of contrail altitudes is valuable for the assessment and improvement of contrail forecasting methods.

Multiple approaches for the retrieval of cloud top altitude using infrared satellite imagery exist. Fundamentally, these retrieval algorithms utilize the fact that the infrared radiance observed by the satellite instrument is a combination of that emitted by the surface, atmosphere and the cloud itself (Liou, 2002). These approaches include radiance fitting (Schmetz et al., 1993), radiance ratioing (Smith et al., 1970), optimal estimation (Heidinger, 2011), and machine learning (Kox et al., 2014; Strandgren et al., 2017a, b). Thin cirrus clouds, which are microphysically similar to aged contrails (Heymsfield et al., 2010), are reported to lead to larger cloud top altitude retrieval errors than other cloud types (Hamann et al., 2014; Chang et al., 2010). Kox et al. (2014) and Strandgren et al. (2017a) address these difficulties by developing the COCS and CiPS neural networks using collocated CALIOP LIDAR and SEVIRI image data, in order to retrieve cirrus properties from measurements by SEVIRI only. The COCS and CiPS algorithms have also been applied to estimate contrail altitude and optical thickness (Vázquez-Navarro et al., 2015; Schumann et al., 2021; Wang et al., 2023). The ability of such cirrus altitude estimation algorithms to estimate contrail altitude has not been tested extensively. Given that younger contrails differ from natural cirrus in their microphysical, optical, and geometrical properties (Kärcher, 2018), such cirrus altitude estimation algorithms may not be appropriate for contrails detected in satellite imagery. To investigate this, we also develop cirrus altitude estimation algorithms in this work and evaluate their performance when applied to contrails.

In this paper, we present an algorithm for estimating the altitude of a contrail by use of GOES-16 ABI infrared satellite imagery. The algorithm is used in combination with the contrail detection method that was introduced by Meijer et al. (2022) to provide three-dimensional estimates of contrail locations above the contiguous United States. To develop this algorithm,

contrails detected in GOES-16 ABI imagery are collocated with measurements by the CALIOP LIDAR. The resulting dataset of 3267 contrails is analyzed for spatio-temporal trends in contrail altitudes, and is compared with flight ADS-B data and measurements by GRUAN radiosondes. The dataset is then used to train a convolutional neural network for the task of contrail altitude estimation algorithm. Similar to Kox et al. (2014) and Strandgren et al. (2017a), we also develop cirrus altitude estimation algorithms using collocated CALIOP L2 and GOES-16 ABI data to investigate their suitability for estimating contrail altitude, and quantify the benefit of training on contrail data specifically. The estimates by the finalized algorithm are compared with flight altitudes derived from ADS-B data. Finally, the algorithm is applied to a full day of data in order to investigate the persistence and evolution of contrail forming regions in 3D.

## 2    Methods

### 2.1    CALIOP and IIR data

The Cloud and Aerosol Lidar with Orthogonal Polarization (CALIOP) is a LIDAR instrument aboard the polar-orbiting CALIPSO satellite (Winker et al., 2010). The CALIPSO satellite was part of the A-train satellite constellation until September 2018, when it was placed in a different orbit together with Cloudsat (Braun et al., 2019). CALIOP measures the attenuated backscatter at the 532 and 1064 nm wavelengths, which are disseminated as the CALIOP L1B product. Additional processing involving the identification of cloud and aerosol layers, as well as extinction retrieval, is performed to create the CALIOP L2 products (Winker et al., 2009). Both the CALIOP L1B and L2 products are used for collocation with GOES-16 ABI imagery.

The CALIPSO satellite carries two other instruments: the Wide Field Camera (WFC) and the Imaging Infrared Radiometer (IIR). The IIR instrument measures infrared radiances at three central wavelengths 8.6 μm, 10.6 μm and 12.0 μm (Garnier et al., 2018). The imagery from the 10.6 μm band and its difference with the 12.0 μm band are used during the collocation process for visualization and manual inspection (see Figure S4 and Figure S5 in the Supplement).

### 2.2    GOES-16 Advanced Baseline Imager

The Advanced Baseline Imager (ABI) is a passive imaging radiometer that is carried aboard the GOES-R satellite series (Kalluri et al., 2018). The ABI features 16 imaging bands, 10 of which are in the infrared part of the electromagnetic spectrum. The nadir resolution of the ABI infrared bands is 2 km, and imagery is available every 5 minutes for the contiguous United States (CONUS) and every 10-15 minutes for the full disk. The contrail detection algorithm by Meijer et al. (2022) uses orthographically projected GOES-16 ABI imagery, centered on CONUS. The boundaries of the domain where contrail detections are available are shown in Figure 2.

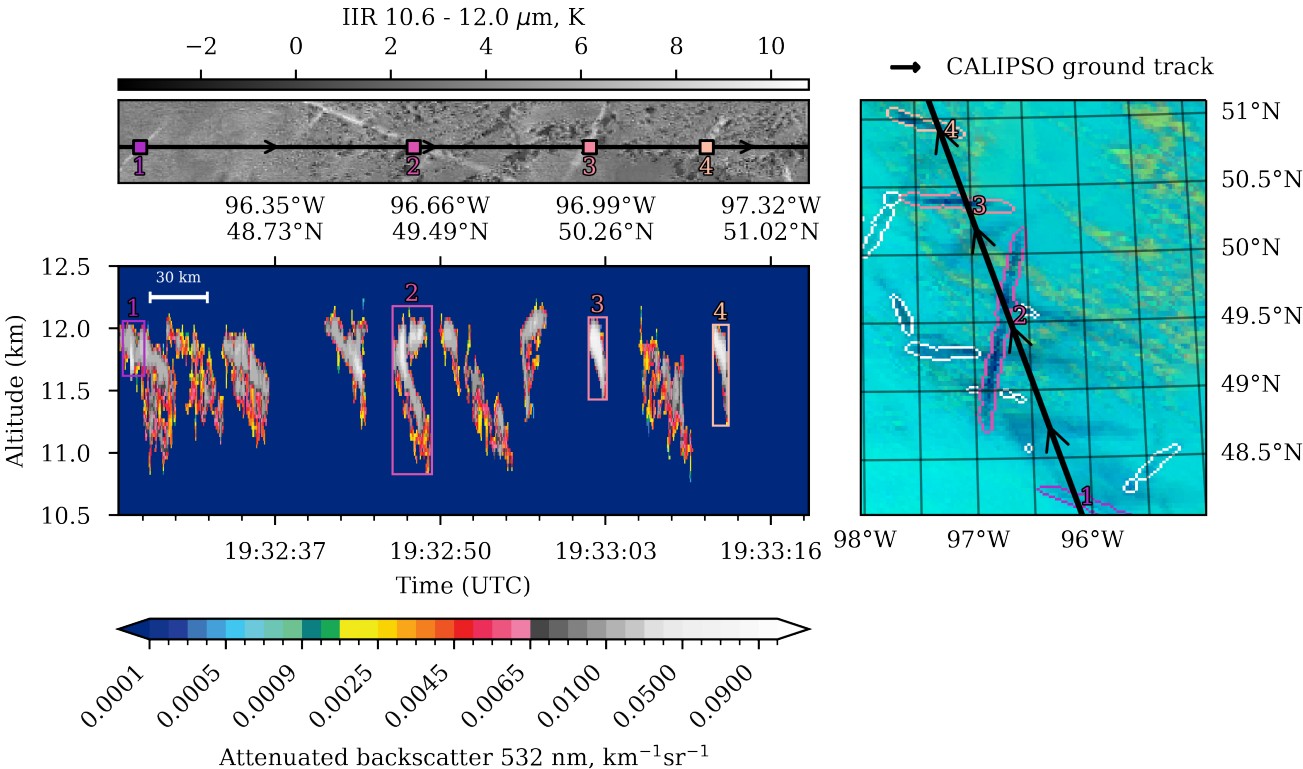

**Figure 1.** An example collocation of contrails detected on GOES-16 ABI imagery (right) in CALIOP L1 data (bottom left). The IIR brightness temperature difference image corresponding to the CALIOP segment is shown as well (top left). The GOES-16 ABI image is a false color ash product, created using the ABI-L2-MCMIPC data for August 8 2018 19:30 UTC. Detected contrails are indicated by their boundaries, which are colored for collocated contrails and white for non-collocated contrails. The corresponding cross-sections in the CALIOP L1B data are shown using bounding boxes, and their approximate location in the IIR image is shown using square markers of the same color. The ground track of the CALIPSO satellite is also shown in the GOES-16 ABI and IIR images, with the arrows indicating the direction of movement. The CALIOP colormap is the one provided by Kuma (2010).

## 2.3 ERA5

The ERA5 reanalysis (Hersbach et al., 2020) is used to provide wind data for the advection performed during the collocation process and comparisons to flight data. Geopotential data is used for conversion between geometric altitude and pressure altitude. The land/sea mask is used as an input for the altitude estimation models.

## 2.4 Collocation

Contrails have previously been located in CALIOP data by use of contrail detections in Aqua MODIS imagery (Iwabuchi et al.,
2012). At the time, the Aqua satellite was in the same orbit as CALIPSO such that the time difference between overpasses by

the two satellites was around 75 seconds. Collocation of contrails detected on imagery of the GOES-16 ABI is different in that the two satellites do not share an orbit. The collocation is performed in two steps. The first "coarse" step, considers whether contrails are detected within a 50 km distance of the ground-track of CALIPSO (see section S1 of the Supplementary Materials for more details). The second "fine" step accounts for advection of contrails between the CALIPSO overpass time and the GOES-16 ABI capture time, as well as parallax displacement. The collocation is performed for the years 2018 to 2022: the resulting dataset features 3267 collocated contrails.

An example collocation is shown in Figure 1, which displays the CALIOP L1B attenuated backscatter and the corresponding IIR brightness temperature difference (BTD) imagery for a segment of an overpass on August 8 2018. The attenuated backscatter is filtered to remove noise using the methods described in section S1. The GOES-16 image shows that the CALIPSO satellite passes over 4 detected contrails, which are indicated using colored boundaries, with their corresponding cross-sections in the CALIOP L1B denoted by bounding boxes (and numbers). The approximate locations of these collocated contrails is also shown in the IIR image in Figure 1. The first contrail encountered in Figure 1 appears to be a thinner and potentially younger contrail embedded in another cloud, and shows up as a narrow region of increased backscatter (white, rather than gray) in the CALIOP L1B data. Contrail 2 has a larger vertical extent (> 1 km) when compared to the other contrail cross-sections found in this segment, and it is also appears to be wider than the other contrails when viewed in the GOES-16 image. This particular contrail may actually be two different contrails given that the structure in bounding box 2 in the CALIOP L1B data shows two backscatter maxima, but these are not discernible as such at the resolution of the GOES-16 image ($\sim 2$ km) or the IIR image (1 km). The increased width of this contrail in the CALIOP data is partially caused by the angle between the CALIPSO ground track and the contrail. When correcting for this angle however, contrail 2 is still found to be wider than contrails 3 and 4. The final two collocated contrails (3 and 4 in Figure 1) are of similar vertical and horizontal extent, both featuring a horizontally displaced fallstreak as a result of vertical wind shear. The GOES-16 ABI image shows several cirrus clouds (such as the one between contrails 2 and 3) that could be contrails that are no longer recognized as such by the detection algorithm, and which also seem to correspond to areas of increased backscatter (in gray) in the CALIOP L1B data. This highlights that, if a different contrail detection algorithm is used, the collocation process could be repeated to find more and/or different contrails.

In addition to locating contrails in CALIOP L1B data, we collocate 5 km cirrus layers - as identified in the CALIOP L2 product - with GOES-16 ABI data. The collocation process is nearly identical as that for contrails, except that the contrail detections described in subsection 2.2 are no longer involved. This "cirrus dataset" is created to develop cirrus altitude estimation algorithms similar to that presented in Strandgren et al. (2017a), but for data from GOES-16 ABI rather than Meteosat SEVIRI. These algorithms are then assessed on their ability to estimate contrail altitudes, and compared to algorithms developed using contrail data specifically. There exists the possibility that contrails are recognized as cirrus layers by the CALIOP processing algorithms and thus appear in the CALIOP L2 5 km cloud layer product. These layers are removed from the cirrus dataset using the dataset of collocated contrails. More details on the collocation processes for both the contrail and cirrus datasets are given in the Supplementary Materials (section S1 and section S2).

## 2.5 Correction of parallax displacement

The parallax displacement refers to the apparent displacement of clouds on satellite imagery that results from the incorrect geolocation of pixels containing the cloud in question. The effect is particularly noticeable at low satellite viewing angles and can amount to a displacement in the order of 10 km over the contiguous United States, when viewed by the GOES-16 ABI. As such, neglecting this effect during the collocation procedure could lead to erroneous results. We correct the parallax effect using Vicente's method (Vicente et al., 2002; Bieliński, 2020).

## 2.6 Advection

As part of the collocation procedure described in subsection 2.4, we account for the advection of contrails by wind. The same advection process is also utilized when comparing flight data to contrail observations. This advection is done by means of the ERA5 data described in subsection 2.3. The ERA5 wind data is interpolated by means of a bicubic interpolation in longitude and latitude (Press et al., 2007), a quadratic polynomial interpolation in pressure, and a linear interpolation in time. The advection process is advanced in time by use of the fourth order adaptive Runge-Kutta method from Fehlberg (1970) implemented in the GNU Scientific Library (Galassi et al., 2002). The vertical position of the contrails/flight segments is affected by the vertical wind speeds from ERA5, but no ice particle sedimentation processes are accounted for.

## 2.7 Conversion between geometric, geopotential and pressure altitudes

The data from the CALIOP instrument uses the altitude above mean sea level (MSL) as its vertical ordinate, thus being a *geometric* altitude. In contrast, most numerical weather prediction and ADS-B data utilizes pressure (or pressure altitude) as vertical coordinates. Therefore, the process of advecting contrails and flight segments, as well as the comparison of collocated contrail altitudes with flight data, requires the conversion between geometric and pressure altitude. This conversion is performed by use of ERA5 geopotential altitude data, which also utilizes the MSL as its datum. We assume that the geopotential altitude is equal to the geometric altitude: this leads to errors at higher altitudes as the the gravitational field of the Earth weakens. Using the fact that the gravitational acceleration is proportional to the inverse square of the distance from the Earth's center, we estimate that this assumption leads to errors of $\sim$10 m when applied to objects around 10 km altitude.

Utilizing this assumption, we find the pressure altitude corresponding to a given object's geometric altitude as follows. We first interpolate the ERA5 geopotential data to the time and horizontal location of the object, at multiple pressure levels. We then estimate the pressure at which the geopotential altitude of ERA5 equals the geometric altitude of the object. This results in an estimated pressure for the object. Finally, to convert this pressure to the pressure altitude, we use the US standard atmosphere (Atmosphere, 1976). The process of converting a pressure (altitude) to a geometric altitude follows these steps in reverse.

## 2.8 GRUAN data

Humidity and temperature profiles measured by radiosondes in the the GCOS Refence Upper Air Network (GRUAN) are used to compare the altitudes of regions conducive to persistent contrail formation with the altitudes found in the collocated

CALIOP contrail dataset (Dirksen et al., 2014). The profiles supplied by GRUAN have already been corrected for radiosonde limitations such as the radiation dry bias and temperature-dependent lag.

We use data from the Lamont (SGP) site as this is located within the region for which contrail detections are available, and it is also has more than 100 Vaisala RS92 sonde launches per year for the period between 2009 and 2018 (Sommer et al., 2016). For the profiles corresponding to these launches, we compute the relative humidity with respect to ice by use of the

approximations from Hyland and Hexler (1983). This relative humidity with respect to ice is used to ascertain whether ice supersaturation (ISS) is present, which is combined with the Schmidt-Appleman criterion (SAC) (Schmidt, 1941; Appleman, 1953) to evaluate whether persistent contrails may form. The SAC is computed by use of the approximation from Schumann (1996), using the parameters shown in Table S2 of the Supplementary Materials.

## 2.9 ADS-B data and comparison of flight and contrail altitudes

Automatic Dependent Surveillance-Broadcast (ADS-B) data is used for comparing observed and predicted contrail altitudes with flight altitudes. We use a mixture of data from the OpenSky database (Schäfer et al., 2014) and FlightAware (for times in 2023). The OpenSky data is used for comparison with the contrail altitude estimation performance on the test set in section 3, whereas the FlightAware data is used for analyzing contrail detections and altitude estimates for a full day of data in section 4. The OpenSky network does not have complete coverage throughout the domain for which we have contrail detections, and we

thus use this data only in areas of the domain where full coverage is available.

When comparing flight and contrail altitudes, we account for the motion of the contrail between the time of formation and the time we observe it in a satellite image by advecting flight segments with ERA5 wind data as described insubsection 2.6. We advect flight data from the 2 hours leading up to the observation of a particular contrail. This choice of advecting only the past 2 hours of flight data is motivated by the results from Chevallier et al. (2023) who show that lifetimes of contrails

observed in GOES-16 ABI imagery tend to be shorter than 2 hours. Only advected flight segments that are sufficiently close - as determined by an estimate of ERA5 wind uncertainty and the duration of the advection process - to the contrail are considered in the comparison. The altitudes of the flight segments that are sufficiently close to the contrail are used to create a distribution of the distance flown at every 10th flight level. Further details of this process can be found in section S9 in the Supplementary material.

## 2.10 Machine learning approaches

We develop both contrail and cirrus altitude estimation techniques using machine learning algorithms. Specifically, we use neural networks (Goodfellow et al., 2016) to "learn" a mapping between the input variables (e.g. GOES-16 ABI infrared brightness temperatures) and the contrail/cirrus top altitude. Neural networks are capable of learning non-linear relationships from data, and have been successfully applied to remote sensing retrieval problems (Kox et al., 2014; Strandgren et al., 2017a;

Amell et al., 2022).

We broadly distinguish between two neural network models: pixel-by-pixel models and image-level models. Pixel-by-pixel models, such as those developed by Strandgren et al. (2017a), operate independently on each individual pixel of the input

satellite image. Image-level models consider (part of) the image simultaneously. Although the pixel-by-pixel option may lead to models containing fewer parameters and better interpretability of the results, image-level models have been shown to perform
better for certain retrieval tasks (Amell et al., 2022). For cloud altitude estimation, it could be that viewing a satellite image (rather than a single pixel) allows a machine learning model to identify cloud overlap, a situation that is known to lead to less accurate predictions by some approaches (Heidinger, 2011), and incorporate this information into the resulting prediction. For the pixel-by-pixel models, we consider multilayer perceptrons (MLP) with varying number and sizes of hidden layers, as well as activation functions. For the image-level model, the convolutional neural network (CNN) from Amell et al. (2022) is used.
More details on the architecture of these neural networks can be found in the Supplementary Materials in section S6.

The pixel-by-pixel models obtain the relevant inputs for a given collocated GOES-16 ABI pixel and are trained to output the CALIOP cloud top altitude that was collocated to the relevant pixel. For the image-level models, we extract patches of 32 by 32 pixels that contain one or more collocated GOES-16 ABI pixels and present these to the CNN, similar to Amell et al. (2022). The CNN predicts the cloud top altitude for all pixels in the input image, but only the pixels in the output image which
have CALIOP collocations are used during training: an example can be found in the Supplementary Materials in Figure S11. When using the CNN for contrail altitude predictions after training is finished, the resulting output (for the whole image) is only considered for pixels where contrails are detected.

We use all 10 of the GOES-16 ABI infrared channels as input, as well as the latitude, satellite viewing zenith angle, and an ERA5 land/sea mask. Temporal information is represented similar to Strandgren et al. (2017a) by two additional inputs
$\sin\left(2\pi\frac{\text{DOY}}{365}\right)$ and $\cos\left(2\pi\frac{\text{DOY}}{365}\right)$ where DOY is the day of the year. The pixel-by-pixel models also receive the regional maximum brightness temperature for all 10 of the ABI channels, also suggested by Strandgren et al. (2017a). This regional maximum is the maximum in a 29 by 29 pixel patch centered on the collocated ABI pixel: this approximately corresponds to the region size used for SEVIRI by Strandgren et al. (2017a). The image-level models also receive a cloud mask as input. For the cirrus dataset, these cloud masks are obtained from the GOES-R ABI cloud top phase (ABI-L2-ACTP) product (Heidinger et al., 2020). In
this mask, the pixels for which the product indicates that the cloud top phase is ice and the data quality flag is "overall good quality" are set to 1, whereas other pixels take on the value of 0. For the contrail dataset, the outputs of the contrail detection algorithm are used as cloud masks. The CNN for contrail altitude estimation is trained by fine-tuning the CNN trained for cirrus altitude estimation on the contrail dataset. This avoids overfitting the CNN, which has significantly more parameters than the MLP models, to the relatively small contrail dataset. The fine-tuning process starts with the CNN trained on cirrus data. Then,
all parameters except for those in the last two layers of the CNN are fixed to the values obtained by training on cirrus data. The parameter values of the last two network layers are optimized by training on the contrail dataset.

With the exception of the ERA5 land/sea mask and some of the inputs to the ACTP processing algorithm (Heidinger et al., 2020), all inputs to the neural networks used for altitude estimation are derived from observed satellite radiances. Previous machine learning approaches for cirrus altitude estimation have made use of NWP data such as surface temperature (Kox et al.,
2014; Strandgren et al., 2017a). Motivated by the potential of using the resulting contrail altitude estimates for the comparison to (and potentially assimilation with) NWP data, we choose not to include these inputs such that the resulting altitude estimates represent an independent piece of information.

For training of the neural networks, 80% of the available data (both for the cirrus and contrail dataset) is used, and the remaining 20% is split evenly between a validation and test set. The validation set is used during training to monitor overfitting,

and the test set is used to evaluate the resulting models. The randomization is performed by grouping collocations by their CALIPSO overpass, and then selecting the overpasses for the train, validation and test sets at random. This avoids placing two or more nearby (or even neighboring) collocated pixels in the training and test sets, which could lead to an unrealistic evaluation of the algorithm's performance on unseen data.

In order to model predictive uncertainty, we use quantile regression neural networks (Pfreundschuh et al., 2018). Rather than

having a neural network output a single estimate of the cloud top altitude, we output an estimate for several quantiles of the cumulative distribution function. These models are trained using the quantile loss function from Pfreundschuh et al. (2018). A known issue with quantile regression neural networks is the phenomenon of quantile crossing, where the "quantiles" predicted by the network are not monotonically increasing. We address this similar to Amell et al. (2022), by fitting an isotonic regression to the neural network outputs. The frequency of occurence and magnitude of these quantile crossings will be discussed in the

Results section.

## 3 Results

### 3.1 Contrail dataset statistics

The dataset consists of 3267 contrails collocated between CALIOP L1 and GOES-16 ABI data for the years 2018 to 2022. The location of all contrails is shown in Figure 2. The mean contrail top altitude decreases with increasing latitude, as was

found by Iwabuchi et al. (2012). The variance of this contrail top altitude distribution is found to increase with latitude. The location of the tropopause is known to decrease, on average, with increasing latitude. Given that ice supersaturation, a necessary condition for contrail persistence, is mostly found below the tropopause (Spichtinger et al., 2003a), this is a likely explanation of the observed decrease in the contrail top altitude with increasing latitude. Further analysis on the location of the collocated contrails relative to the local tropopause can be found in section S5 in the Supplementary Materials. We find that the majority

(86.6%) of collocated contrails are located below the local tropopause.

In Figure 3 we show the seasonal distribution of the contrail top altitudes within the dataset, accompanied by vertical distributions for persistent contrail formation conditions and flight traffic. The data for persistent contrail formation is obtained from GRUAN profiles from the SGP site (location shown in the inset in Figure 3). The pressure altitude, which is used as the vertical coordinate in Figure 3, is defined as the altitude in the International Standard Atmosphere corresponding to the

pressure at which the contrail top is found. Since aircraft use pressure altitudes for vertical navigation, some seasonal variation in geometric contrail top altitudes (as measured by CALIOP) is expected to be caused by changes in temperature and thus the geometric altitude of certain pressure levels. To exclude this seasonal variation in the geometric altitude of a particular pressure level, Figure 3 uses the pressure altitude as its vertical ordinate. Finally, the flight altitude distributions in Figure 3 result from 1000 randomly sampled hours of ADS-B data in the years 2018 and 2019. Given the different locations and times at which the

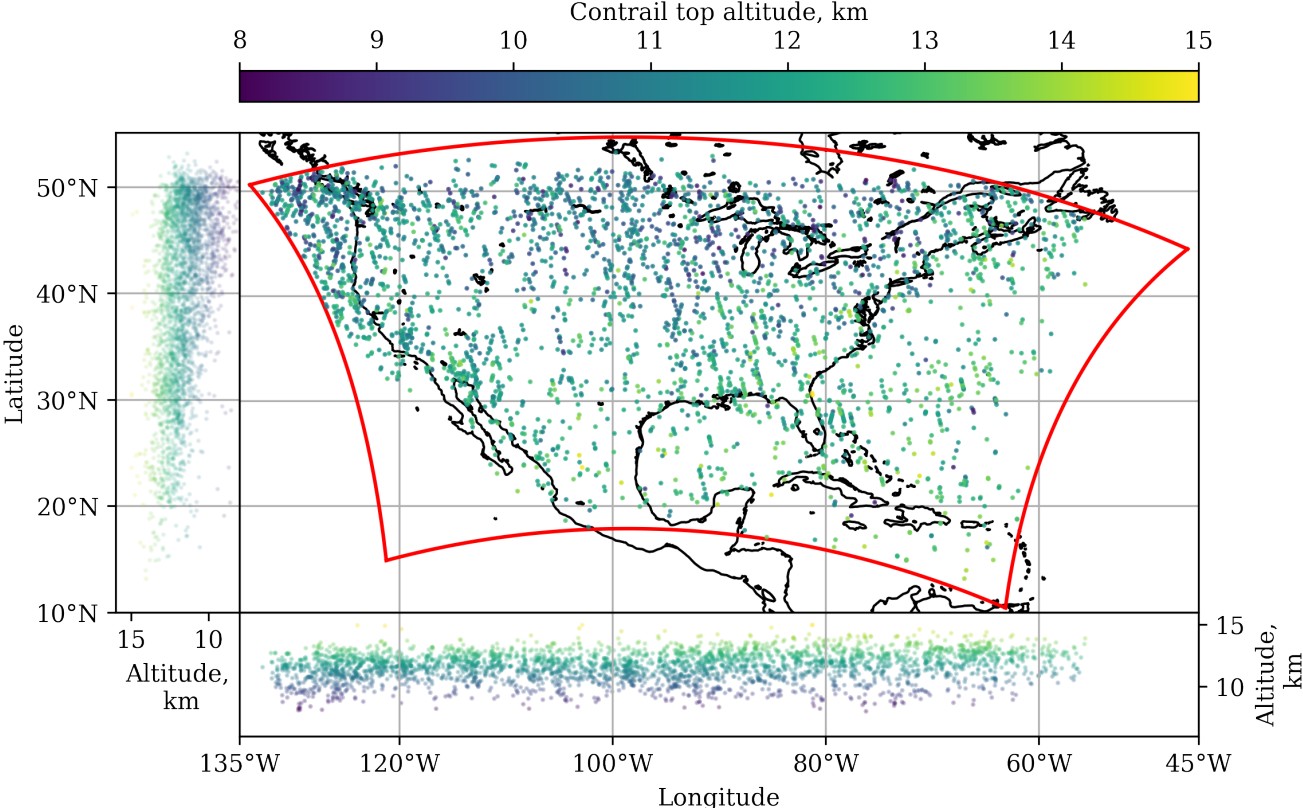

**Figure 2.** Location of the 3267 contrails that were collocated between CALIOP and GOES-16 ABI data, over the years 2018 to 2022. The red line indicates the boundary of the domain for which contrail detections are available. The smaller plots on the side of the main plot show the variation of contrail top altitude with longitude and latitude. All altitudes in this figure refer to the geometric top altitude of the contrails.

data in the three different plots is collected, Figure 3 serves mostly to compare seasonal trends in the altitude of flights, regions conducive to persistent contrail formation, and observed contrail top altitudes.

On average, contrails are located at a pressure altitude of 11.1 km during winter (DJF) and 11.8 km during summer (JJA). This 700 m seasonal difference in the mean contrail location is statistically significant using Welch's T-test (Welch, 1947) at $p = 0.01$. In terms of geometric altitude, the difference between winter (11.3 km mean) and summer (12.2 km mean) is larger

at 0.9 km than the pressure altitude difference, at the same level of statistical significance. The vertical distribution of regions satisfying SAC and ISS (middle plot in Figure 3) also features higher altitudes during summer and lower altitudes in winter. Such seasonal changes in the vertical distribution of ice supersaturation have been reported by other studies (Spichtinger et al., 2003a; Treffeisen et al., 2007; Gierens et al., 2020b) that use radiosondes, also finding that ice supersaturation is on average located at higher altitudes during Northern Hemisphere spring/summer than in Northern Hemisphere winter. The contrail top

altitudes in Figure 3 are generally found to be higher than the regions that are ice supersaturated and satisfy the Schmidt-

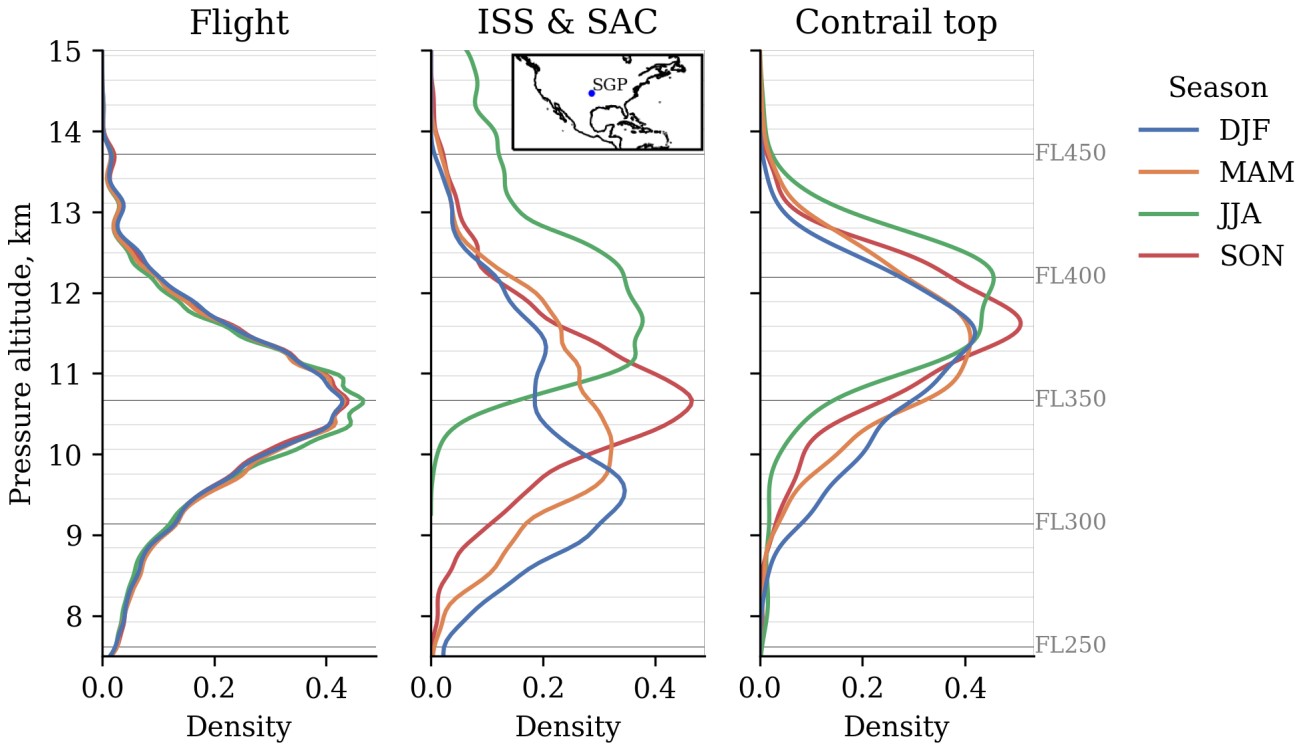

**Figure 3.** Kernel density estimates for the vertical distribution of contrail top altitude, persistent contrail forming regions and flight traffic as a function of season. Each curve integrates to 1. The horizontal gray lines show flight levels. The persistent contrail forming regions are defined as ice supersaturated regions (ISS) that satisfy the Schmidt-Appleman criterion (SAC). The flight distribution is weighted by the distance flown at each altitude, and is based on OpenSky ADS-B data from 1000 randomly sampled hours in the years 2018 and 2019. The ISS & SAC data is obtained from GRUAN profiles at the SGP site, whose location is indicated in the inset. The GRUAN data covers the years 2009 to 2021.

Appleman criterion. Given the limited spatial coverage of the GRUAN data as compared to the contrail top altitudes (whose spatial distribution was shown in Figure 2) as well as the different times at which this data was captured, this discrepancy is likely due to sampling effects.

The flight altitude distributions in Figure 3 show that flight distance peaks at flight levels 340 - 360 (10.5 to 11 km). Seasonal changes in the weighted means of these distributions are all smaller than 100 meters. Thus, the vertical variation of the observed contrail top altitude distributions cannot be explained by seasonal patterns in flight altitude alone: consideration of the seasonal variation in the location of persistent contrail formation conditions is necessary. This seasonal variation in the vertical location of persistent contrail forming conditions is likely due to seasonal patterns in tropopause pressure, as ISSRs are known to be located closely below the tropopause (Spichtinger et al., 2003a; Petzold et al., 2020).


The single GRUAN site for which data is shown in Figure 3 does not provide full coverage of the spatial domain in which contrails are collocated (see Figure 2), nor does it have a significant amount of sonde launches for the period 2018-2022 (31 versus 4486 launches between 2009 and 2017). Although statistics on the vertical distribution of ISS are known to vary with location and time (Spichtinger et al., 2003a, b; Treffeisen et al., 2007; Lamquin et al., 2012; Gierens et al., 2020b), Petzold et al. (2020) analyzed 15 years of aircraft in situ measurements and did not find any long-term trends in the occurrence of ISS besides

seasonal variations. Similarly, we find that for the 9 years between 2009 and 2018 the seasonal trends in the altitude of regions satisfying both the SAC and ISS measured at the SGP site are qualitatively consistent (see Figure S8 in the Supplementary Materials).

## 3.2  Algorithm performance

This section evaluates the performance of the four machine learning approaches to contrail altitude estimation that were de-

scribed in subsection 2.10, by comparing the predicted altitudes to those measured by CALIOP for a subset of data that was not used during training (i.e. the test set). The evaluation of the predicted probability distributions for contrail top altitude is the subject of subsection 3.3: this section focuses only on the means of these distributions.

The performance of the four different altitude estimation models on the contrail test dataset is shown in Figure 4. These four models differ in the dataset(s) they have been trained (cirrus, contrails or both) on and whether they consider pixels individually

(MLP) or not (CNN). The cirrus altitude estimation models (Cirrus MLP and Cirrus CNN) are included in this comparison in order to evaluate their suitability for contrail altitude estimation, as well as the benefits of training an altitude estimation model on contrail data rather than cirrus data only. The two plots on the left of Figure 4 show the mean errors as a function of altitude as a percentage of the ground truth altitude. For all altitudes, the two MLP models show larger absolute errors than the two CNN models. The two MLP models also show larger relative errors for low (< 10 km) and high (> 14 km) contrails. The mean

error (rather than the mean absolute error) plot in Figure 4 illustrates that the MLP models have the tendency to over-predict the altitude for "low contrails", and under-predict the altitude for "high contrails". The mean errors for the two CNN models are more uniform with altitude. The cirrus CNN achieves lower errors for the lower and upper ends of the altitude range considered here (8 - 15 km) than the contrail CNN, but the opposite is true for the altitudes between 10 and 12 km.

The scatter plots on the right of Figure 4 illustrate the spread between the ground truth (CALIOP altitudes) and the model

estimates. We see that three out of four models achieve a root mean square error (RMSE) smaller than 1 km, with the exception being the Cirrus MLP (which operates pixel-by-pixel). This particular model, which is trained using cirrus data only, performs worse on the contrail test set (RMSE of 2.87 km) than on a dataset of unseen cirrus data points (RMSE of 1.05 km, shown in section S7 of the Supplementary material). This Cirrus MLP model is similar in design and performance to that introduced by Strandgren et al. (2017b), for which a detailed comparison can be found in section S7 of the Supplementary material.

The RMSE of a model can also affected by the dataset it is evaluated on. For example, a model that always over-predicts by 10% would have a higher RMSE on a dataset that has more variance. For this reason, we also show the coefficient of determination $R^2$ in the scatter plots in Figure 4, which takes such dependency on the underlying data into account. For the Cirrus MLP evaluated on the contrail test set, the $R^2$ is -0.39. This can be interpreted as the model being worse than using the

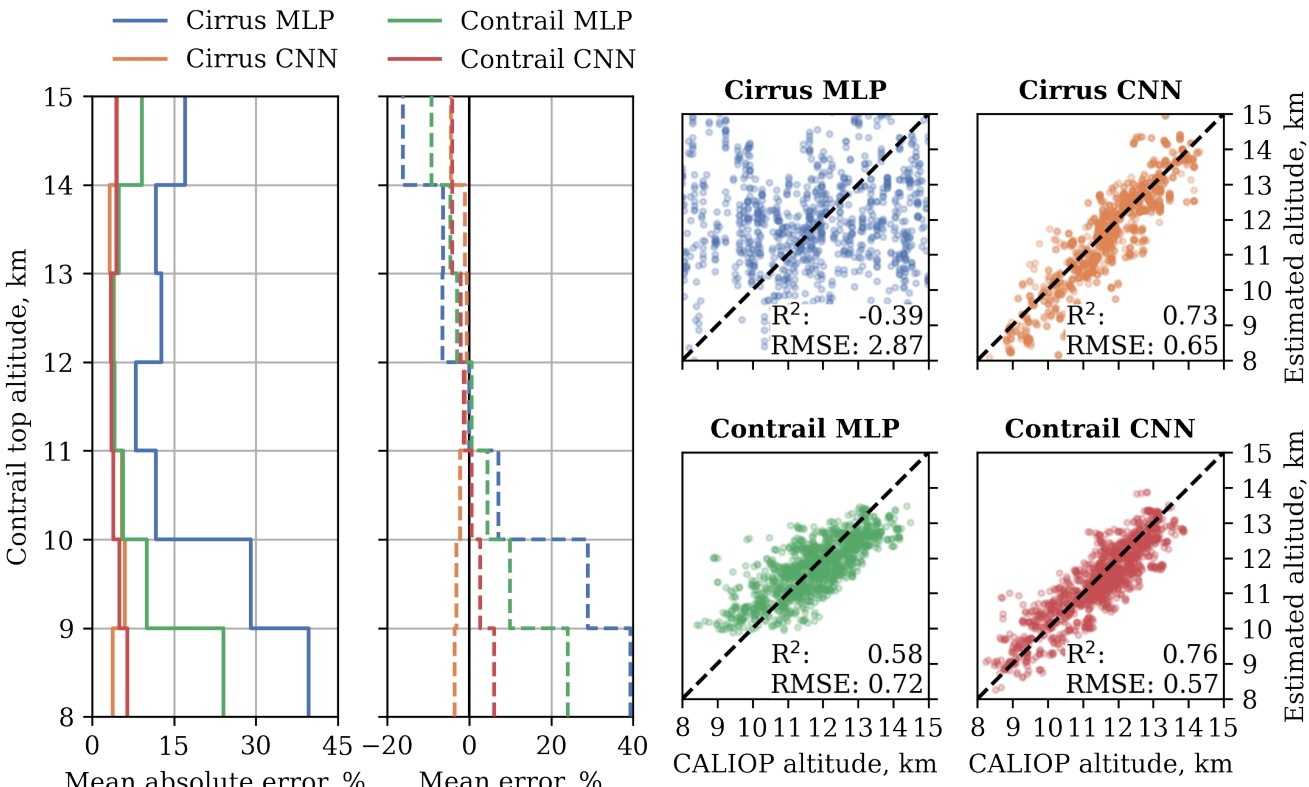

**Figure 4.** Performance of the pixel-by-pixel (MLP) and image-based (CNN) models on the contrail test dataset. The left two plots show the relative and absolute errors as a function of the contrail top altitude. The scatter plots on the right show the estimates by the algorithms for all data points within the test set, as well as the coefficient of determination $R^2$ and the root mean square error in km. Each point corresponds to a collocated GOES-16 ABI pixel. All altitudes in this figure are geometric ones.

mean of the evaluation data as a predictor. However, when evaluated on unseen cirrus data point we find a $R^2$ of 0.77, indicating that the Cirrus MLP does have "skill" for the prediction of cirrus altitude estimation. We thus see that the performance of the Cirrus MLP - a cirrus altitude estimation algorithm that operates on a pixel-by-pixel basis - on the task of contrail altitude estimation is lower than what one would expect based on an evaluation with cirrus data alone. Nonetheless, the results of the other three algorithms indicate that this "skill gap" can be resolved by a choice of architecture, training data, or a combination of these.

For example, Cirrus CNN algorithm which operates on multiple pixels simultaneously and was trained with cirrus data only, is found to predict contrail altitudes *with* skill ($R^2$ of 0.76). Its skill, as measured by $R^2$, is however higher on a set of unseen cirrus data points (0.86) than it is for contrails. These results do suggest that choosing a CNN model architecture over a MLP

is beneficial for both the performance on cirrus and contrail data. Moreover, the Cirrus CNN is found to perform better in the task of contrail altitude estimation than the Contrail MLP: an algorithm that was trained using contrail data.

Referring to Figure 4, the best performing algorithm is found to be the Contrail CNN. This algorithm corresponds to the Cirrus CNN model that was fine-tuned on the contrail dataset. It combines the benefits of the choice of architecture (CNN over MLP) and the exposure to contrail data to achieve the lowest RMSE of all four algorithms at 570 m. We therefore utilize the contrail CNN for the remainder of this work.

### 3.3 Evaluation of predictive probability distributions

The analysis in the previous section compared the mean contrail top altitudes estimated by the neural networks with the corresponding measurements by CALIOP. The best-performing algorithm, the CNN trained first on cirrus and then fine-tuned on contrail data (red in Figure 4), outputs 13 different quantile values that can be utilized to construct a probability distribution for the contrail top altitude. The estimated quantiles are 0.025, 0.05, 0.1, 0.2, . . . , 0.9, 0.95, and 0.975. This section discusses and quantifies the quality of the distributions constructed using these quantile predictions.

We compare the probability distributions from the CNN with two "baseline" models: one based on the CALIOP dataset distribution (a climatology, in a sense) and one based on the CNN mean and the error distribution on the test dataset. The first of these models represents the situation a priori, without any altitude estimation algorithm, where all that is available is the altitude data of the collocated contrails. The second model does include an altitude estimation algorithm (as it uses the mean from the CNN output) but without predictive uncertainty which depends on the algorithm inputs: i.e. it is constant. An example

of the probability density functions (PDFs) for the three models, and the corresponding contrail altitude found by CALIOP, is shown in the left plot in Figure 5. More examples can be found in Figure S14 in the Supplementary Materials.

  One desired property of predictive probability distributions is that they are *calibrated*. This means that when the model says an event should occur with probability 0.2, the event actually occurs 20% of the time. We assess this, for each predicted quantile, by computing the number of times the CALIOP contrail top altitude is actually below that altitude. As an example,

for the quantile value predicted for the 50% quantile (i.e. the median), we would expect 50% of the CALIOP data points to be below that value if the model is calibrated. By quantifying the fraction of contrails that are below the altitude predicted for each of the quantiles, the curves shown in the right plot of Figure 5 result. The closer a particular curve is to the black dashed line, the better the calibration of the model. Figure 5 shows that the best calibrated model is the one based on the CALIOP dataset distribution, which is expected given that it represents the distribution of the data directly. The CNN tends to be overconfident

for most predicted probabilities, with largest deviations occurring for predictions between probability 0.4 and 0.9. For example, when the CNN predicts that 60% of the contrails should be below a particular altitude, Figure 5 indicates that only 50% of contrails will actually be found below this altitude. This over-confidence can be corrected by means of post-processing: we will show in the next section that the 95% confidence intervals derived from the CNN predictions are calibrated, so we do not perform this correction here. Lastly, Figure 5 shows that the error statistics model is over-confident for the majority of predicted

probabilities, and is the least calibrated model considered here.

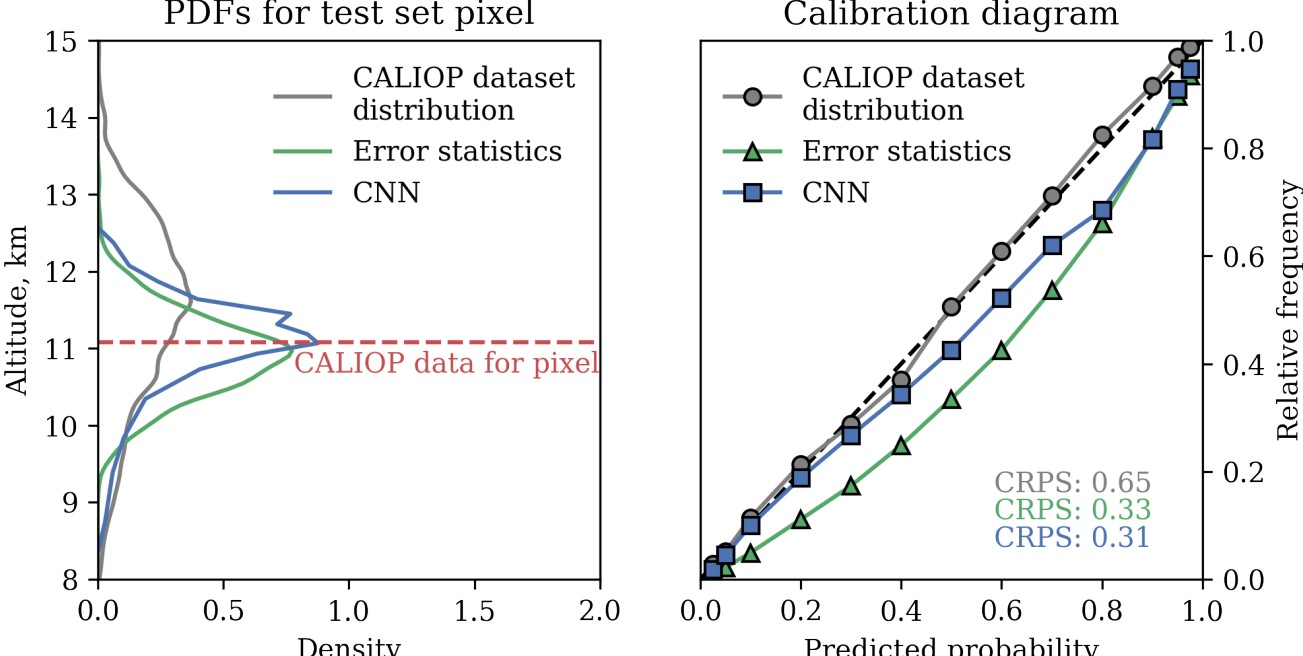

**Figure 5.** (Left) Example probability density functions (PDFs) for a collocated GOES-16 ABI pixel from the test dataset, with altitude referring to geometric altitude. (Right) Calibration diagram for the three different models, obtained using all data in the test set. The CRPS values for each of the three models are also shown. The colors of the lines in both plots, and their meaning, are identical.

If a predictive probability distribution is calibrated, we would also like it to be *sharp*. That is, most of its probability mass should be close to the actual observed value (i.e. the CALIOP top altitude). To assess this, we use the Continuous Ranked Probability Score (CRPS) (Wilks, 2011). For a given probability distribution $F(y) = \mathbb{P}(Y \leq y)$ and observed value $x$, the CRPS is given by the integral

$$\text{CRPS}(F, x) = \int_{-\infty}^{\infty} \left[ F(y) - \mathbb{I}(y > x) \right]^2 dy, \quad (1)$$

where $\mathbb{I}$ is the indicator function. Note that $\text{CRPS}(F, x) \geq 0$, and $\text{CRPS}(F, x) = 0$ if $F(y) = \mathbb{I}(y > x)$ which corresponds to a situation where all probability mass is located at the measurement location $x$. Lower values of the CRPS correspond to "sharper" probability distributions centered on the measurement $x$, and are thus desirable. The mean CRPS values obtained for the three models are shown in Figure 5. The CRPS of the CNN (0.31) and the error statistics model (0.33) are both about two times as low as that of the dataset distribution model (0.65), indicating that the distributions predicted by these models are sharper when compared to the measurements. The improvement of the CNN CRPS with respect to that of the error statistics model is not as large: the CNN CRPS is about 6% lower. In section S8 we further analyze the size of the 95% confidence intervals of the three models and find that the CNN confidence intervals are smaller than that from the error statistics model

for over half the test set data points. Thus, the input-dependent predictive uncertainty of the CNN does provide additional value when compared to the constant uncertainty approach from the error statistics model. The issue of quantile crossing was

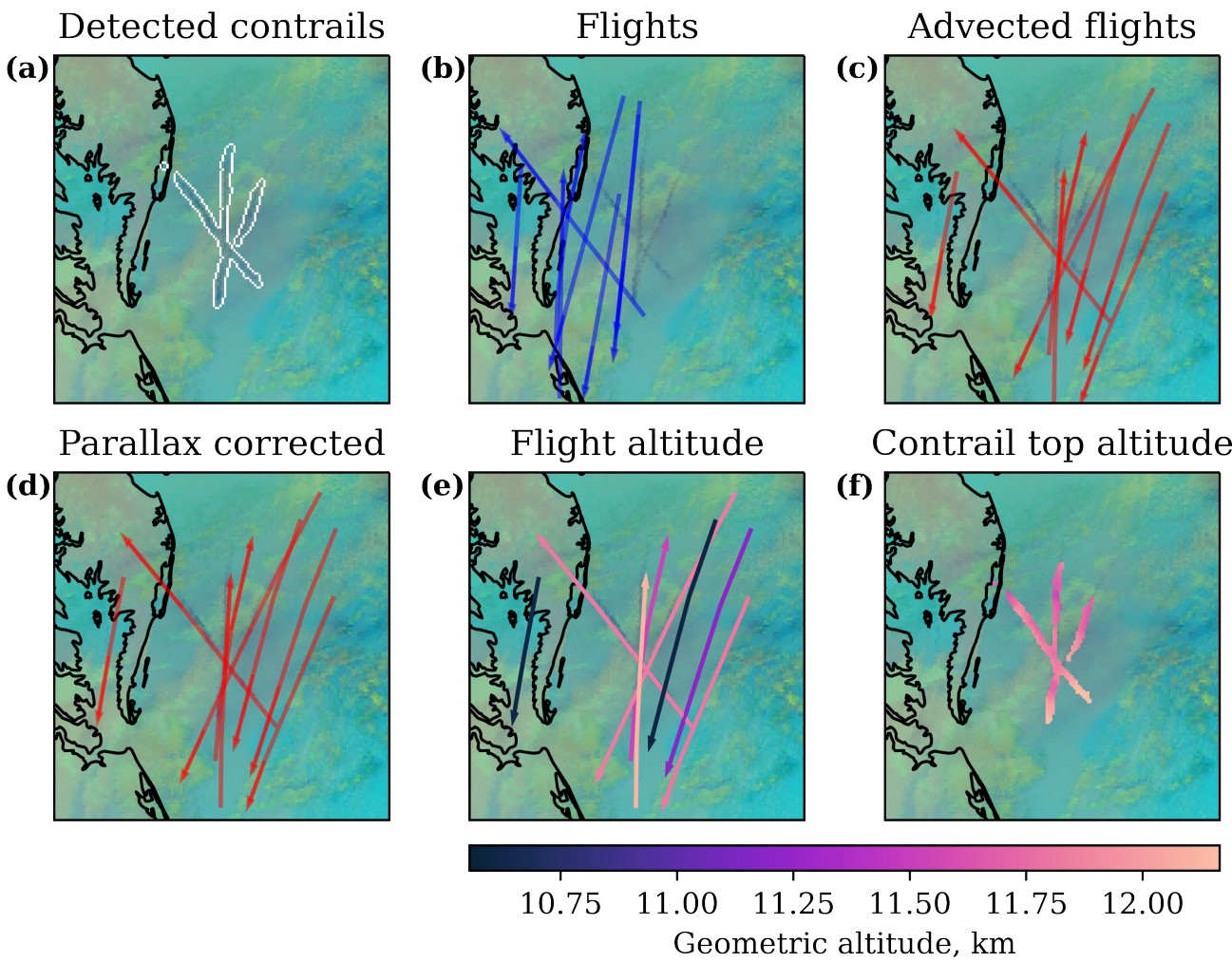

**Figure 6.** Steps involved in comparison of estimated contrail top altitudes with flight altitudes, overlayed on an ash image for GOES-16 ABI-L2-MCMIPF product for 08:10 UTC April 29, 2019. Flight directions are shown using arrowheads. The same colormap is used for subfigures e) and f).


introduced in the methods section and occurs when the quantile values that the neural network outputs are not monotonically increasing, and therefore do not define a valid probability distribution. In the test set, we found 7 (out of 1344 total GOES-16 ABI pixels) occurrences corresponding to a rate of 0.52%. The amount by which the quantiles cross is a distance of 38 meters on average with a maximum of 69 meters: these values represent 0.1% deviations from the values for which predictions are

made (order of 10 km). These quantile crossings are corrected by means of an isotonic regression: the effect on the mean of the distributions is in the order of 10s of micrometers.

## 3.4    Comparison with flight data

This section further evaluates the performance of the contrail altitude estimation algorithm by comparison with flight data. Although interactions with wing tip vortices, wind-driven ascent/descent, radiative heating/cooling and ice particle sedimentation

can play a role, the observed contrail's vertical location is expected to be influenced most strongly by the altitude of the flight that formed it. Thus, the altitudes of flights near an observed contrail can be used to assess the correctness and plausibility of a contrail altitude estimate. Before any such comparison, flight tracks are advected using ERA5 (Hersbach et al., 2020) wind data to account for contrail movement after formation. Only in locations and during times of low air traffic is it plausible to

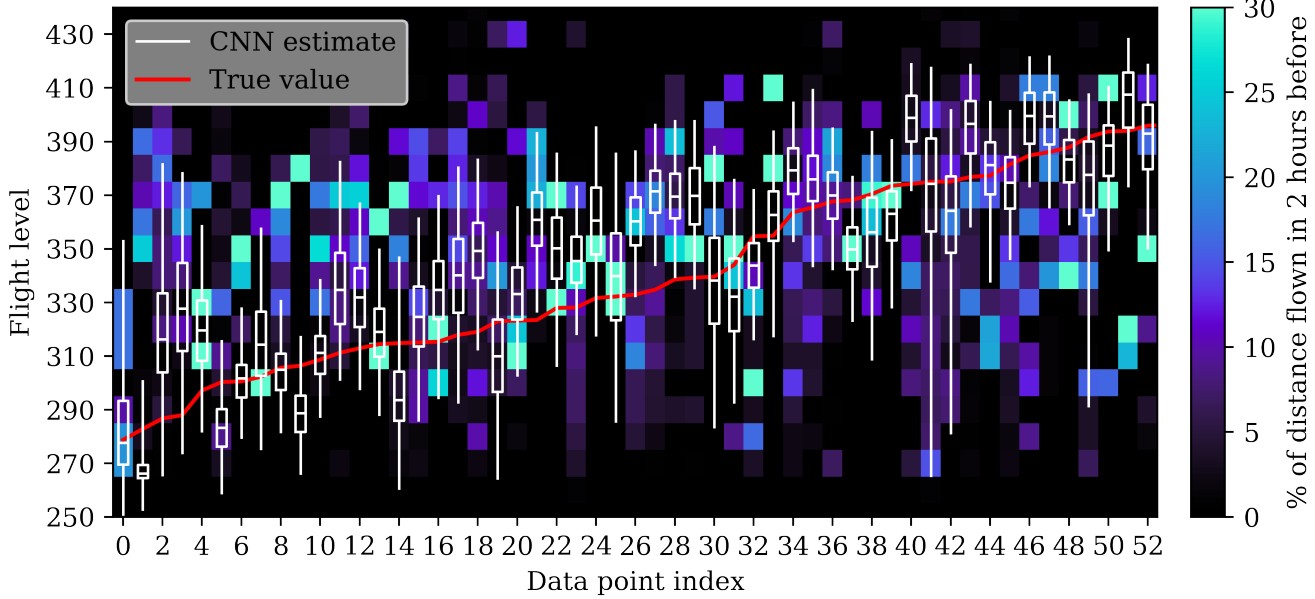

**Figure 7.** Flight altitude and contrail top altitude distributions for samples from the test set. Each column represents a different sample from the test set, including the distribution of distance flown per flight level in the 2 hours leading up to the observation of the contrail. The boxplots show the contrail top altitude distribution estimated by the model, with the box covering the 25th to 75th quantiles. The whiskers cover the 95% confidence interval. The red line connects the contrail top altitude values obtained from the CALIOP data.

visually compare observed contrails and their estimated altitudes with advected flight tracks. An example of such a situation is

shown in Figure 6, depicting a scene off the East coast of the United States at 4 AM local time in the morning. Flight data from the two hours leading up to the image was advected to the satellite image observation time. The flight pressure altitudes are converted to geometric altitudes by use of ERA5 geopotential data. We also apply a parallax correction to "simulate" where the flight tracks would have been visible when viewed by the GOES-16 ABI, which has the effect of further aligning some of the

advected flight tracks with the detected contrails. Figure 6e and Figure 6f show the altitudes of the advected flight tracks and those estimated for the contrails. For the contrail top altitudes, the mean of the distribution estimated by the CNN is shown. The flights best aligning with the 3 detected contrails were cruising at flight levels 380 and 390, which corresponds to geometric altitudes between 11.7 and 12.1 km. The mean of the estimated contrail top altitude of all detected contrail pixels is 11.9 km.

The comparison of flights to contrails is more complicated in areas of higher traffic density, and makes analyses such as that presented in Figure 6 infeasible given currently available tools. For these cases, we compare estimated contrail top altitudes to the distribution of distance flown, rather than individual flight tracks and their altitudes, as shown in Figure 7. The rationale for using these distributions of distance flown is that, in the absence of knowledge on the location of ice supersaturated regions, they can be used to form an "a priori" confidence interval of the vertical location of the contrail. Figure 7 shows that all but 4 contrails (i.e. 7.5%) are found within the 95% confidence interval predicted by the model: which shows that these intervals are close to calibrated. Furthermore, the intervals that capture 95% of distance flown (not shown) are larger than the 95% confidence intervals for the contrail top altitude distribution for all these data points: using the contrail top altitude distributions from the model narrows down the 95% confidence interval of contrail altitude from 3.3 km to 2.2 km, on average.

## 4 A day of 3D contrail coverage over the United States

The previously discussed analyses of the contrail altitude estimation algorithm's performance have mostly relied on comparing its outputs directly with CALIOP data. As CALIPSO (the satellite equipped with the CALIOP instrument) is in a sun-synchronous orbit, contrail cross-sections are only available for certain hours of the day. This prohibits the assessment of the algorithm's performance using CALIOP data throughout the entire day. Instead, we apply the contrail altitude estimation algorithm to 24 hours of contrail detection data between 08:00 UTC on 21 and 22 August 2023, in order to study the temporal consistency and plausibility of the estimated contrail altitudes. The contrail detections are produced by the algorithm presented in Meijer et al. (2022).

The 3D contrail location estimates are visualized in Figure 8, with each sub-plot showing 3 hours of data. Another view of the data, in the form of a time series of contrail coverage and the relative proportion per altitude band can be found in Figure 9. For both figures, the mean of the distribution predicted by the algorithm is shown.

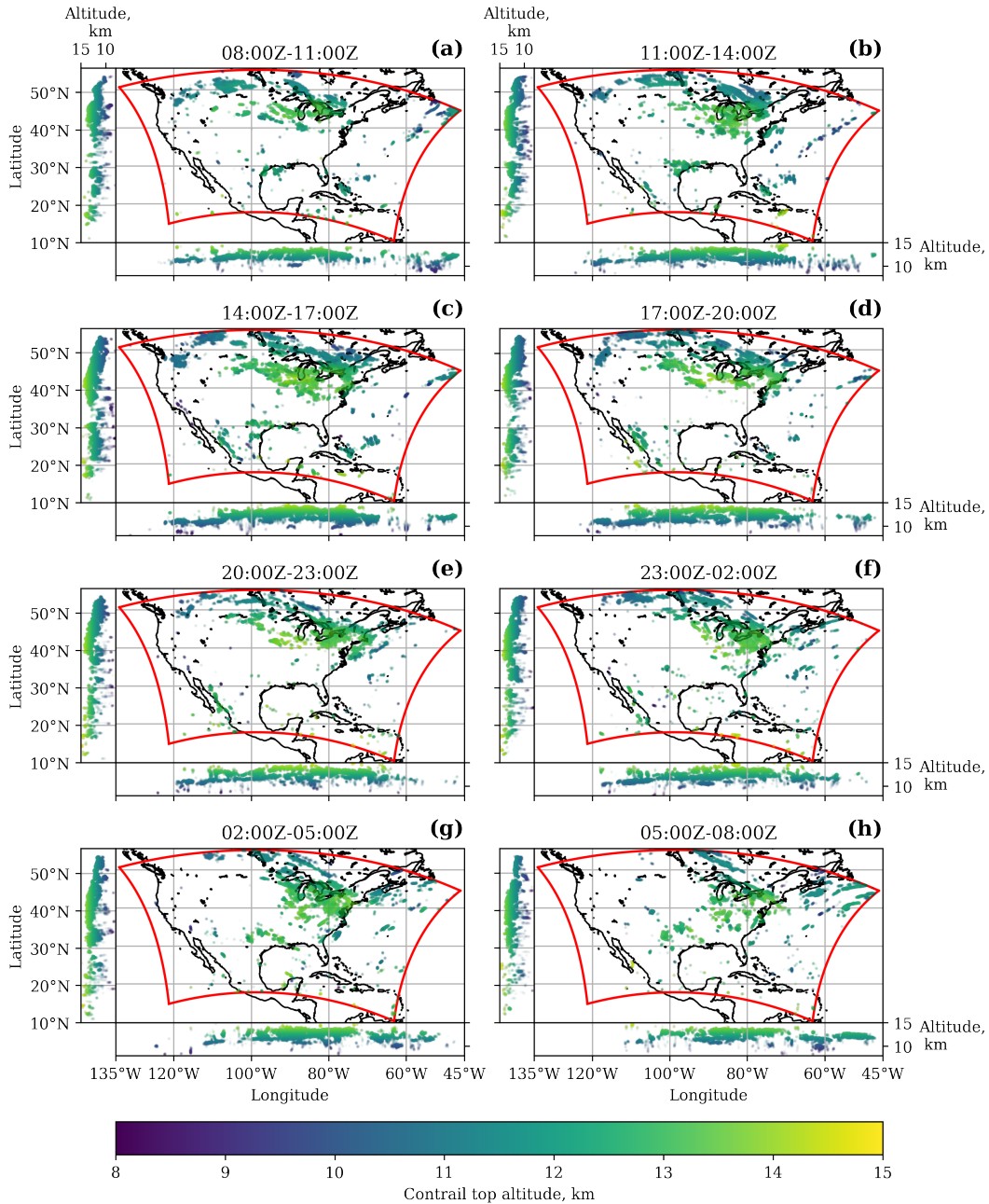

**Figure 8.** Contrail detections and altitude estimates for 24 hours starting from 08:00 UTC on August 21 2023. Each subplot shows 3 hours of detected contrails and estimated altitudes, corresponding to the analysis of 18 ABI-L2-MCMIPF images. The smaller plots on the side of the main plot show the variation of estimated contrail altitudes with longitude and latitude. Each dot corresponds to a detected contrail pixel. All altitudes in this figure are geometric ones.

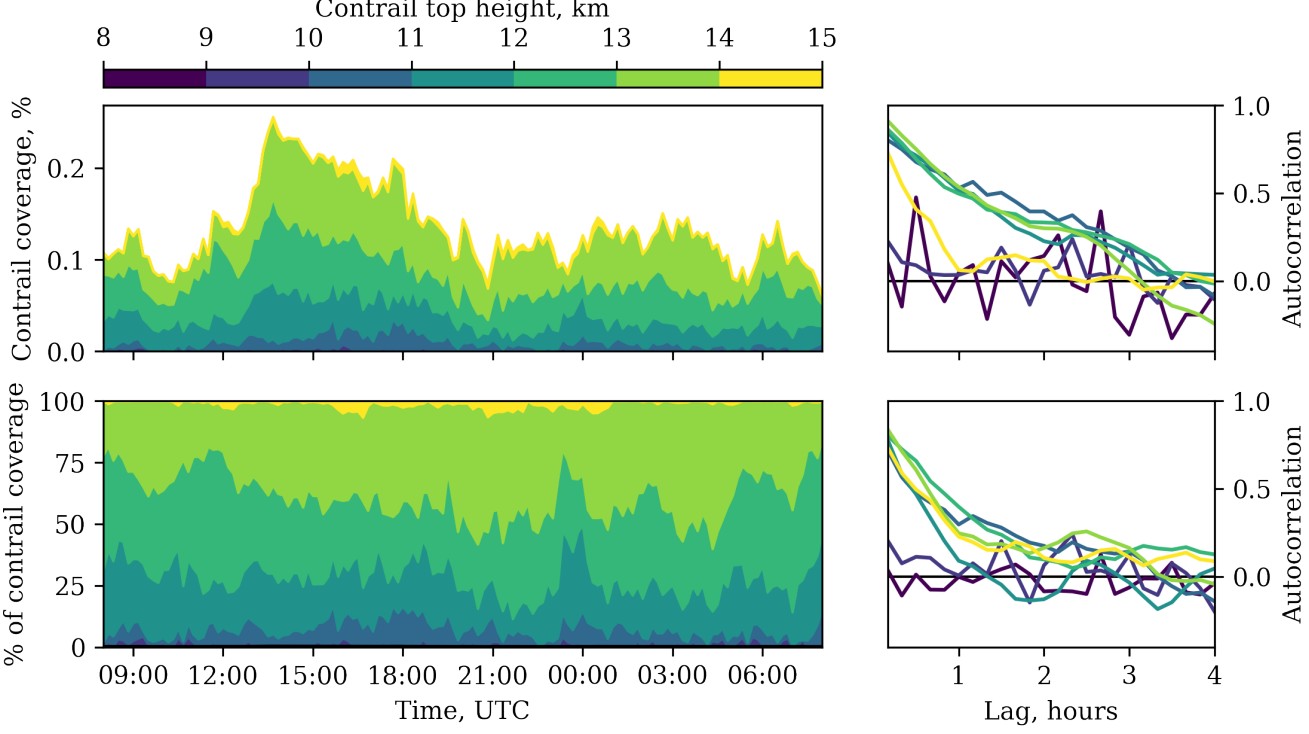

**Figure 9.** Contrail coverage for 24 hours starting from 08:00 UTC on August 21 2023, as well as the contribution per altitude band. Contrail coverage is here defined as the percentage of pixels in the orthographic projection of the ABI-L2-MCMIPF image that are detected as contrails. The plots on the right show the autocorrelation of the time series as a function of time scale ("Lag"). The colors of the curves in the autocorrelation plots correspond to the colorbar of the time series. All altitudes in this figure are geometric ones.

Contrails are detected and vertically located at altitudes between 8 and 15 km throughout the day. The greatest fraction however is found at altitudes between 12 and 14 km, which is consistent with the seasonal distributions presented earlier in Figure 3. The zonal and meridional views in Figure 8 indicate that clusters of contrails are located at similar altitudes: with cluster thicknesses ranging between 500 - 1500 m. Such vertical extents are in line with those observed for regions of ice supersaturation (Spichtinger et al., 2003a). Given that mostly westerly winds were present at cruise altitudes for the day analyzed, the meridional view can be used to track individual contrails and clusters. One example is found between 11:00 and 14:00 UTC (Figure 8b) near 90°W and 55°N at an altitude of about 10 km: the contrail slightly descends as it advects towards the East. The altitude estimates also show consistency throughout the day. For example, a region of contrail formation and persistence is present near the Great Lakes which moves towards the South East during the day, located behind an advancing surface cold front. The contrails present within this region are consistently estimated to be at altitudes between 12.5 and 14 km, which can be best seen in the zonal views. Given that this particular region is present during the majority of the day, it explains the large contribution of contrails between 13 and 14 km as visible in the time series in Figure 9.

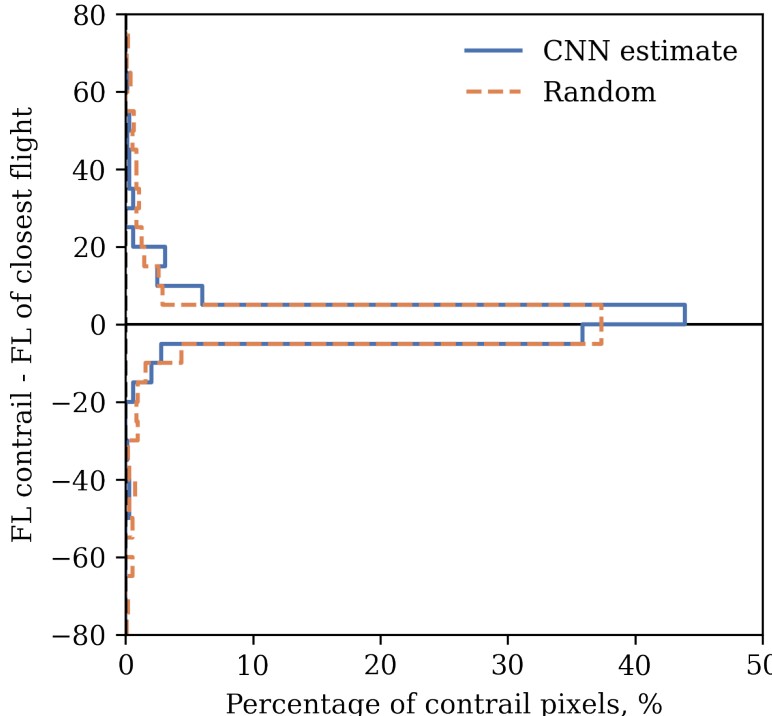

**Figure 10.** Distance in flight level between estimated contrail altitudes and nearby flights, for 10,000 randomly sampled contrail pixels during the 24 hours following 08:00 UTC August 21 2023. For the flight data, the altitudes correspond to those found after advecting the originally flown waypoints.

The time series shown in Figure 9 depict the diurnal variation in contrail coverage and the relative contribution by each altitude band. Contrail coverage peaks around 15:00Z, corresponding to the early morning rise in air traffic around 9-10 AM local time, as found by Meijer et al. (2022). The autocorrelation plots included in Figure 9 show that the contrail coverage within particular altitude bands has autocorrelation coefficients larger than 0.5 at time scales of 1 to 2 hours, but decrease to small absolute values beyond 2 hours. This is likely a consequence of typical detectable contrail lifetimes, which tend to be shorter

than 2 hours for methods that use geostationary satellite imagery (Chevallier et al., 2023). However, the contributions by the lowest altitude band covering 8 to 9 km in altitude oscillates at short time scales (as is evident from the negative autocorrelation values). This is likely a non-physical phenomenon that arises from false positive contrail detections. For example, consider the South West of the United States in the geographic overview for 20:00 to 23:00 UTC in Figure 8e. The few, low altitude contrail pixels in that region are only detected on one or two images within the 3 hours. Because there are relatively few actual contrails

estimated to be within the 8-9 km altitude band, the effect of these false positives is enlarged and oscillatory behavior results. For the other altitude bands, the contributions by actual contrails seem large enough to outweigh this behavior.

The geographic overviews in Figure 8 and the time series in Figure 9 illustrate the regional and temporal behavior of the altitude estimates. In order to assess the plausibility of the contrail altitudes predicted by the algorithm, we compare these with advected flight data similar to the analysis in subsection 3.4. Flight traffic from the 2 hours before each ABI-L2-MCMIPF product time is advected to the observation time to compare flight altitudes with those of nearby contrail pixels. In order to assess the altitude estimate for a given contrail pixel, we compute the distance from each advected waypoint to the location of the pixel. We make the assumption that the error of the wind field used to advect the flight data is at most 10 m/s: if a waypoint has been advected over a time period $\Delta t$ (in seconds), we include it in the analysis for a contrail pixel if it is within a distance of $10 \times \Delta t$ m. The advected flight tracks that satisfy this constraint are used to construct a distribution of distance flown per flight level. Further details of this procedure can be found in section S9 in the Supplementary Materials.

We construct flight altitude distributions according to this approach, for 10,000 randomly sampled contrail pixels from the 24 hours of data. For each of these distributions we find the flight level with non-zero distance flown that is closest to the mean flight level of the contrail as estimated by the algorithm. We then consider the difference between this "closest flight" and the estimated contrail altitude, resulting in the statistics shown in Figure 10. We find that the greatest majority of estimated contrail altitudes are within 20 flight levels (corresponding to 2000 ft, or about 600 m) of at least one nearby advected flight. However, flight altitude distributions tend to be densely populated (note the lack of large "gaps" in the distributions in Figure 7) and thus it is plausible that randomly selected contrail altitudes would lead to similar statistics. To analyze this, a random flight level is sampled uniformly for each contrail pixel (and thus flight distribution) and shown in Figure 10 as well. Comparing the error distributions of the randomly picked and the actual estimated contrail altitudes shows that the latter are closer to an advected flight track: on average, the closest advected flight track for the estimated contrail altitudes is 4.9 flight levels away, whereas this is 9.2 flight levels for the randomly selected contrail altitude. The altitude estimates by the CNN - when compared to the altitude of the closest flight - do show a positive bias of 2.4 flight levels (statistically significant as determined using a one-sample T-test at $p = 0.01$). The evaluation with CALIOP test data indicates no statistically significant bias (at $p = 0.01$) for the CNN estimates, however. Potential other causes of the bias observed in Figure 10 may be the use of geopotential data for the conversion between geometric- and pressure altitudes, the methodology used for constructing the flight altitude distributions, as well as the omission of contrail physics in the advection process.

## 5    Conclusions

We have described the development and performance of a contrail altitude estimation algorithm that uses infrared radiances measured by the geostationary GOES-16 satellite. The developed algorithm can augment existing contrail detection methods to estimate the 3D location of contrails based on satellite imagery, which further enables comparison of observations with models.

A dataset of contrails detected on GOES-16 ABI imagery collocated with CALIOP LIDAR data has been used to develop this altitude estimation algorithm. This dataset of 3267 contrails illustrates the existence of seasonal patterns in the vertical location of contrails, with contrails found at higher altitudes during summer and lower altitudes during winter. Part of this seasonality

is explained by the higher temperatures during summer, which cause pressure levels to be located at a geometrically higher altitude as the air expands. We also found a seasonal trend in the contrail top pressure, which we hypothesize to be due to seasonal differences in the altitude of ice supersaturated layers that satisfy the Schmidt-Appleman criterion.

The contrail altitude estimation has been performed using neural networks. Models that consider a single satellite image pixel at a time, as well as those that ingest an entire image at once, were developed. The image-based models were found to
470 perform better, reaching a root mean square error of 570 meters on the test dataset compared to 720 meters by the pixel-by-pixel model. We have also shown that the probability distributions that are predicted by the algorithm can be used to construct confidence intervals that are calibrated and can vertically locate contrails with greater accuracy than flight data alone.

The spatial and temporal behavior of the contrail altitude estimates have been analyzed by considering 24 hours of contrail coverage above the contiguous United States. The resulting estimates of 3D contrail locations show the evolution of individual
contrail forming regions with thicknesses of 500 - 1500 m, consistent with prior research. One such region, located behind an advancing surface cold front, was observed to move from the Great Lakes towards the East coast during the day. Contrails located within this region were consistently estimated to be at altitudes between 12.5 and 14 km. The plausibility of the estimated contrail altitudes has been assessed by comparison with altitudes of advected flight traffic. On average, an advected flight track from the 2 hours before the observation was found at a vertical distance of 4.9 flight levels from the estimated
contrail altitude. For randomly selected contrail altitudes, this distance was 9.2 flight levels on average.

This work also represents the first remote-sensing based cloud altitude estimation algorithm developed specifically for contrails. We have shown that altitude estimation algorithms developed using cirrus data only, when applied to estimate contrail altitudes, show lower skill than the algorithms that were trained with contrail data. The cirrus pixel-by-pixel model, which achieved a 1.04 km root mean square error on the cirrus test set, showed degraded performance on the contrail test set with an
485 RMSE of 2.87 km. The image-level model, which performed better than the pixel-by-pixel model on the cirrus test set with an RMSE of 0.81 km, achieved an RMSE of 0.65 km on the contrail test set. This is still larger than the image-based altitude estimation model that was developed using contrail data, which has an RMSE of 570 km.

The altitude estimates can be used to compare satellite-based observations of contrails with predictions by models in greater detail than was previously possible, as a vertical comparison is now possible as well. Especially in the context of contrail avoid-
490 ance, a potentially cost-effective and near-term mitigation option for aviation's non-$CO_2$ climate impact, accurate predictions of altitudes conducive to persistent contrail formation are necessary and will benefit from this new observational approach. Additionally, we expect that the dataset of contrails collocated in CALIOP LIDAR data developed as part of this work will prove to be useful for a detailed assessment and potentially calibration of contrail models.

*Code and data availability.* The collocated dataset, code for collocation and algorithm development, and the developed deep learning models
will be made publicly available upon acceptance of the manuscript.

*Author contributions.* VRM performed all analysis. SDE, IAW and SRHB supervised the study. All authors contributed to the preparation of the manuscript.

*Competing interests.* The authors declare no competing interests.

*Acknowledgements.* The CALIOP data was obtained from the NASA Langley Research Center Atmospheric Science Data Center. Hersbach, H. et al., (2020) was downloaded from the Copernicus Climate Change Service (C3S) (2023). We would like to thank Dr. Martin Schäfer from the University of Oxford for granting us access to the OpenSky database archive, and Labelbox for providing us with a platform to label our data. We thank Dr. Johan Strandgren for answering any questions related to his work. This research was supported by NASA research grant 80NSSC19K0943 and by the U.S. Federal Aviation Administration Office of Environment and Energy through ASCENT, the FAA Center of Excellence for Alternative Jet Fuels and the Environment, project 78 through FAA Award Number: 13-C-AJFE-MIT under the supervision of Nicole Didyk Wells. We would like to thank Ziming Wang and an anonymous referee for their reviews, which we believe to have significantly improved this manuscript.

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
