# Peer review of "Contrail altitude estimation using GOES-16 ABI data and deep learning"

_EGUsphere, 2024_

## Author Comment (AC1)

**Review 1**

**This paper presents the first remote-sensing-based contrail altitude estimation algorithm. Both the image-level model and the cirrus pixel-by-pixel model are developed and compared, with an evaluation of predictive uncertainty and an assessment of the method's accuracy using individual test data and independent flight data. This study offers valuable insights for further assessing the climate impact of contrail cirrus. The paper is well-organized and well-written. I urge its publication in AMT, with some minor comments provided for the authors' consideration.**

We thank the reviewer for their positive evaluation of the paper, and the several specific comments that helped improve the manuscript. The responses to the specific comments are shown below.

**Specific comments:**

**Line 40: Please provide the physical explanations for why the infrared channels are used for estimating cloud top altitude.**

We have added the sentence "Fundamentally, these retrieval algorithms utilize the fact that the infrared radiance observed by the satellite instrument is a combination of that emitted by the surface, atmosphere and the cloud itself (Liou, 2002).".

**Figure 7: The plot shows a trend where the CNN generally overestimates contrail altitude compared to the true values from CALIPSO. Are there any potential ideas for this?**

We have computed the bias, defined as

$$\text{bias} = \frac{1}{n} \sum_{i=1}^{n} (\hat{y}_i - y_i),$$

where $n$ is the number of data points, $\hat{y}_i$ is the mean of the probability distribution output by the CNN for data point $i$ and $y_i$ is the CALIOP value for data point $i$.

Although the subset of data shown in Figure 7 does have a positive bias of 2.86 flight levels (87 meters), an analysis on the full dataset shows no statistically-significant bias at the p<0.05 level. This suggests that the bias shown in Figure 7 is a result of the specific circumstances of those contrails, rather than a systematic bias in the algorithm.

**Figure 10: The plot here seems to support my impression from Figure 7 that the contrail altitude can be slightly overestimated. During data collocation, you carefully**

**considered the advection of aircraft data due to horizontal wind. Then, contrail ice crystals can sediment, which should theoretically reduce the altitude rather than increase it when compared to the flight data. Are there any reasons behind this discrepancy?**

This is an interesting point. We think that any discrepancy between the estimated contrail top altitude (in flight levels) and selected "closest" flight can be due to a combination of the following:

- Estimation error by the algorithm
- Conversion error from geometric altitude to pressure altitude, owing to errors in the used geopotential data
- Incorrectly choosing the "closest" flight (i.e. we do not compare the estimated altitude with the flight that actually formed the given contrail)
- Dynamical effects: contrail sinking + deepening due to the aircraft wake, formation of the secondary wake, buoyancy effects, radiative heating/cooling, gravitational settling of contrail ice particles, vertical winds and more.

Firstly, we do indeed find that the bias (i.e. the mean of the blue distribution shown in Figure 10) is non-zero (and positive) at a statistical significance level of $p = 0.01$. The analysis presented in response to the previous comment showed that it cannot be concluded at a statistically significant level (with any $p$-value lower than 0.29) that the model has a bias. However, this analysis pertains to the entirety of the test set, which contains data points that were randomly picked from all available data spanning the years 2018 to 2022. There does exist the possibility that particular circumstances (combination of season and synoptic conditions) lead to a positive or negative bias in the algorithm's estimates when averaged over a 24 hour period. However, the test data stratified by season does not indicate any seasonal biases at a statistically significant level. Assessment of the impact of specific synoptic conditions on model performance would require more analysis.

The other possible sources of discrepancies between the "closest flight" altitude and the estimated contrail top altitude may also lead to biases of either sign. The reviewer is indeed correct in noting that gravitational settling of ice particles, which is not included in the advection of the flight data, would lead to lower "advected" flight altitudes and would therefore increase the positive bias observed here.

Summarizing, there is indeed a bias present in Figure 10. However, the objective of the analysis presented therein is to show an agreement between the CNN estimates and flight altitudes that could potentially have formed the contrail. Given the lack of ground truth data on which flight formed the contrails whose altitudes are estimated that day, it is not

possible to assess the relative importance of possible contributions (i.e. CNN error, geopotential error, etc.) to the bias. This also drove the decision to perform most of the quantification of the altitude estimation algorithm's performance with CALIOP data, rather than using results from existing flight-to-contrail matching approaches.

We have added the following sentences to the manuscript to reflect the above analysis:

*"The altitude estimates by the CNN - when compared to the altitude of the closest flight - do show a positive bias of 2.4 flight levels (statistically significant as determined using a one- sample T-test at p = 0.01). The evaluation with CALIOP test data indicates no statistically significant bias (at p = 0.01) for the CNN estimates, however. Potential other causes of the bias observed in Figure 10 may be the use of geopotential data for the conversion between geometric- and pressure altitudes, the methodology used for constructing the flight altitude distributions, as well as the omission of contrail physics in the advection process."*

**Conclusion: The RMSE is used as the metric to indicate the accuracy of the algorithm, as emphasized in the abstract. Since the developed contrail altitude retrieval method is the next step due to the biased prediction of ice supersaturation vertical extension in contrail avoidance, would it be better to also show the simple mean bias error or mean absolute error for estimating the contrail altitude?**

When comparing the RMSE and the mean absolute error (MAE), the RMSE is more sensitive to outliers than the MAE. We evaluated the MAE for the four different models and found the values to always be lower than the corresponding RMSE values. The conclusions on the relative performance of the four models is the same as when using the RMSE. The results from the mean bias error (which we have used in the response to a previous comment, but simply called it "bias") for the CNN have been discussed already. For the other three models, the mean bias error is found to not be zero (with a maximum mean bias error of 540 meters for the Cirrus MLP). Given the role of the RMSE in the evaluation of the probabilistic component of the CNN (to construct the predictive model with constant uncertainty) and the identical conclusions achieved when using the MAE, we choose to leave the latter metric outside of our consideration.

**Technical corrections:**

**Caption of Figure 1: "Zulu" time is equivalent to "UTC" time. However, I'm not sure if it is widely used in this research field. This applies to the entire text to be consistent with the figure.**

We have reviewed the submission guidelines set by the journal and have found that the correct way to indicate this is indeed by use of the "UTC" abbreviation. We have modified this throughout the manuscript.

**L90: "a 50km distance of the ground-track of CALIPSO." I assume it should refer to the supplement S1.**

We agree that it is helps the reader to refer to the supplementary materials here, in particular to the section mentioned by the reviewer. Thank you! We have added the following to this sentence: *"(see section S1 of the Supplementary Materials for more details)."*

**L132: "FlightAware (for times in 2023)". Eventually it appears not to have been used because the focus was on the years 2018-2022.**

The FlightAware data has been used in the analysis discussed in Section 4 of the manuscript, as this concerns a day of data analyzed in 2023. However, we realize now that the description of where the two different data sources are used was lacking, so we have added the following sentence to the manuscript.

*"The OpenSky data is used for comparison with the contrail altitude estimation performance on the test set in section 3, whereas the FlightAware data is used for analyzing contrail detections and altitude estimates for a full day of data in section 4."*

**L221: "ISS" instead of "ISSRs".**

We think that the use of both ISS and ISSRs is possible here.

**L273: tends to be over-confident for probabilities between 0.5 and 0.9, as well as between 0.1 and 0.2.**

We have updated the manuscript to more accurately reflect the results shown in Figure 5. We thank the reviewer for noticing this error.

*"The CNN tends to be overconfident for most predicted probabilities, with largest deviations occurring for predictions between probability 0.4 and 0.9. For example, when the CNN predicts that 60% of the contrails should be below a particular altitude, Figure 5 indicates that only 50% of contrails will actually be found below this altitude."*

**Overall, the excellent work presented in this article is acknowledged.**

We again thank the reviewer for their positive evaluation of our manuscript.

---

## Author Comment (AC2)

**Review 2**

**Summary:**

**The authors use two deep learning techniques to estimate contrail cloud top height in GOES infrared imagery trained with CALIOP lidar data. Over 3000 contrails over five-year period are collocated with the lidar data to allow for the building and testing of the deep learning methods. The more successful contrail height estimate method was developed from a convolutional neural network (CNN), estimating contrail height with a root mean square error of 570 m in test data. An analysis of the CNN method results show that the predictive probability of the CNN method is generally well calibrated and has a smaller 95% confidence interval than the confidence intervals derived from flight altitude data alone. The authors also processed a 24-hour period of GOES data to show the spatial and temporal distribution of the contrail height estimates.**

**General comments:**

**The overall quality of the manuscript is good. The authors explain the methodology and results of the research concisely, and reach logical and consistent conclusions.**

We thank the author for the time taken to review the manuscript and the comments provided.

**Although the topic of the manuscript is contrail height estimation, the authors include discussion of thin cirrus height estimation that is unnecessary (considering the title of the manuscript) and confusing (especially subsection 3.2). Unless the authors can show why the cirrus altitude estimation is integral to the research presented in the main manuscript, I suggest that discussion about cirrus altitude estimation be removed from the paper.**

We think it is essential to include this discussion for the purposes of this manuscript. One of the core objectives of the paper is to develop a contrail altitude estimation algorithm, but we consider it crucial to motivate this objective by showing that existing approaches (similar to what we refer to as "Cirrus MLP") for cirrus height estimation – which have been applied to contrails as well – do not perform as well as expected. Moreover, the development of the "Cirrus CNN" algorithm is essential for the development of the "Contrail CNN" algorithm. This is because the Contrail CNN is constructed by taking the Cirrus CNN and fine-tuning it on the dataset of collocated contrails. In turn, the Cirrus CNN has been developed using collocations of cirrus clouds in CALIOP L2 data with the corresponding GOES-16 ABI infrared radiances. Hence, omission of a discussion of thin cirrus height estimation would not make sense from this perspective either.

With this in mind, we realize that the manuscript as previously written was not making this point adequately. We have added a statement in lines 52-53 of the introduction which highlights the reason for investigating the use of cirrus altitude estimation models for contrails. We also reiterate this point when introducing our cirrus-trained algorithm, and when comparing its performance to that of the contrail-trained algorithms (lines 290-292). Finally, in the conclusions we use our evaluations of the improved performance of the contrail-trained algorithm to argue that future research should not rely on approaches which were trained only on cirrus data, while also recommending that multi-pixel (CNN) methods can outperform single-pixel methods (lines 483-484).

**The authors make multiple references to the Supplementary Materials, so much so that it is nearly impossible to understand the manuscript without also reading those pages. As a result, the manuscript is incomplete and might not stand alone. The reader should not have to rely on the Supplementary Materials to read the principal paper. Finally, the paper lacks references to multiple concepts that should be explained in the paper (not just the Supplementary Materials). I can find no references for the various height conversions (between geometric, geopotential, and pressure altitudes), the advection of contrails, or the parallax correction used in the main paper. Add these references to the manuscript.**

The comments by the reviewer on the large number of references to the Supplementary Materials are appreciated and have been addressed. We have expanded the discussion in the manuscript on the following topics by use of material found in the Supplementary Materials:

- Construction of flight altitude distributions (section 2.9)

Furthermore, we have added new material on the following topics to the main paper:

- Conversion between geometric, geopotential and pressure altitudes (section 2.7)
- Advection (section 2.6)
- Parallax correction (section 2.5)

**Specific comments:**

**Lines 100-104: Could not the width of contrail 2 also be the result of the geometry of the contrail relative to the CALIPSO ground track? Most of the other contrails are nearly perpendicular to the ground track, while the angle between contrail 2 and the satellite track is much more acute?**

Part of the increased width (as viewed in the CALIOP data) is due to the angle this particular contrail makes w.r.t. the CALIPSO ground track. However, we think that this alone is not

enough to explain the relatively large width of the contrail, compared to contrails 3 and 4. We have added the following sentences to the manuscript:

*"The increased width of this contrail in the CALIOP data is partially caused by the angle between the CALIPSO ground track and the contrail. When correcting for this angle however, contrail 2 is still found to be wider than contrails 3 and 4."*

**Lines 109-117: This paragraph is unclear.. The authors state "The collocation process is nearly identical at that for contrails, except that the contrail detection masks are no longer involved." I can't find any mention of cloud masks up to this point.**

We have modified the sentence to:

*"The collocation process is nearly identical as that for contrails, except that the contrail detections described in subsection 2.2 are no longer involved."*

**Line 156: What are "normal" operations? Even after reading the paper, it is not clear to me what that means.**

We have modified the sentence to:

*"When using the CNN for contrail altitude predictions after training is finished, the resulting output (for the whole image) is only considered for pixels where contrails are detected."*

**Lines 172-173: '…all inputs to the neural networks are "observational"…' What is "observational" in this context? Why not say instead that the inputs are derived from the satellite radiances alone with no additional (NWP data) used?**

Indeed, the wording suggested by the reviewer here is more exact. We have modified the sentence to:

*"With the exception of the ERA5 land/sea mask and some of the inputs to the ACTP processing algorithm (Heidinger et al., 2020), all inputs to the neural networks used for altitude estimation are derived from observed satellite radiances."*

**Figure 2 and elsewhere: It is not always clear which altitude the authors are using. In Figure 2, for example, what type of altitudes are plotted here? Geometric? Geopotential? Pressure altitudes?**

We have modified the manuscript in the following places to clarify which altitude is referred to:

- Figure 2
- Figure 4
- Figure 5

- Figure 8
- Figure 9

**Lines 194-195: What is the "thickness" of the altitude distribution? It appears in Figure 2 that the variance of the altitude distribution increases as the latitude increases, contrary to the text.**

We sincerely thank the reviewer for noticing this error. Indeed, the reviewer is correct in that the variance of the altitude distribution increases as the latitude increases. We have modified the text to read:

*"The mean contrail top altitude decreases with increasing latitude, as was found by Iwabuchi et al. (2012). The variance of this contrail top altitude distribution is found to increase with latitude."*

**Figure 3: Why are the contrail tops generally so much higher than the ISS & SAC regions (except for summer)?**

We think this is mostly due to sampling effects, given the limited spatial extent of the GRUAN data at the SGP site and the amount of temporal overlap between the ISS & SAC measurements and the contrail altitude data points. We have added a sentence on this to the manuscript:

*"The contrail top altitudes in Figure 3 are generally found to be higher than the regions that are ice supersaturated and satisfy the Schmidt-Appleman criterion. Given the limited spatial coverage of the GRUAN data as compared to the contrail top altitudes (whose spatial distribution was shown in Figure 2) as well as the different times at which this data was captured, this discrepancy is likely due to sampling effects."*

Earlier in the discussion of figure 3, we have also added a sentence that emphasizes the utility of the data in figure 3 to be mostly to compare seasonal trends:

*"Finally, the flight altitude distributions in Figure 3 result from 1000 randomly sampled hours of ADS-B data in the years 2018 and 2019. Given the different locations and times at which the data in the three different plots is collected, Figure 3 serves mostly to compare seasonal trends in the altitude of flights, regions conducive to persistent contrail formation, and observed contrail top altitudes."*

**It is apparent from the paper that the three profiles (Flight, ISS & SAC, Contrail top) represent entirely different times, locations, and number of observations. It would be better to make this distinction much more clear to the reader, otherwise they may be confused by this figure.**

We agree with the reviewer that this might be confusing. We have revised Figure 3 to include an inset that shows the location of the GRUAN site from which data is used, and have modified the caption to point out that the SAC & ISS data is from a single GRUAN site. The caption now also repeats the point about the flight data being from randomly sampled times in the years 2018 and 2019. In the response to the previous comment, we have also added another sentence to the manuscript that re-iterates this point.

**Section 3.2: The authors refer much more to the Supplementary Materials here than the manuscript itself. Many of the values stated in the text don't match any of the values presented in Figure 4. This is very confusing! As stated earlier, I suggest the authors remove any discussion of cirrus altitude from the paper. It is superfluous and not presented well.**

We agree with the reviewer that the discussion of Figure 4 in the original manuscript could be improved significantly. We have therefore made the following changes:

- Figure 4 itself has been revised to more clearly indicate the correspondence between the scatter plots and the models they are for, as well as more descriptive axis labels.
- We have revised subsection 3.2 completely. We have added an introductory paragraph that we hope clarifies the context of this subsection and its relation to the rest of the manuscript. We have also repeated some of the details of the four different altitude estimation models, and have expanded on the discussion of Figure 4.

**Figure 5: Green line, blue line, black line. Which models do they represent? A legend would make this figure much easier to understand.**

Our intention was to avoid visual clutter by using the same color scheme between the two plots in this figure (and hence the legend in the plot on the right also applied to the one in the left). To avoid any such future confusion, we have added the same legend to the plot on the left. We thank the reviewer for pointing out this source of confusion.

**Line 311: Why is "simulate" in quotation marks? It appears to be a simulation (*i.e,* it imitates the appearance of) in the true sense of the word. The parallax correction is actually making the alignment of the flight tracks match better with the detected contrails.**

This is a good point: we have removed the quotation marks.

**Figure 7: Unless they looked that the Supplementary Materials, the reader would not know what "% of distance flown in 2 hours before" would mean. Some description of**

**how this quantity was obtained must be included in the manuscript, not just the Supplementary Materials.**

We completely agree with the reviewer here and we apologize for the confusion. We have modified the manuscript as follows:

- Under methods, the section on "ADS-B data" has been extended with a discussion of the methodology used to obtain these quantities.
- In the paragraph starting with *"The comparison of flights to contrails is more complicated in areas of higher traffic density, and makes analyses such as that presented in Figure 6 infeasible given currently available tools. For these cases, we compare estimated contrail top altitudes to the distribution of distance flown, rather than individual flight tracks and their altitudes, as shown in Figure 7."*, we have removed the reference to the supplementary material.